# Choice, Motives, and Mixed Messages: A Qualitative Photo-Based Inquiry of Parents’ Perceptions of Food and Beverage Marketing to Children in Sport and Recreation Facilities

**DOI:** 10.3390/ijerph19052592

**Published:** 2022-02-23

**Authors:** Rachel Prowse, Kate Storey, Dana Lee Olstad, Valerie Carson, Kim D. Raine

**Affiliations:** 1Faculty of Medicine, Memorial University of Newfoundland, St. John’s, NL A1B 3V6, Canada; 2School of Public Health, University of Alberta, Edmonton, AB T6G 1C9, Canada; kate.storey@ualberta.ca (K.S.); kim.raine@ualberta.ca (K.D.R.); 3Cumming School of Medicine, University of Calgary, Calgary, AB T2N 4N1, Canada; dana.olstad@ucalgary.ca; 4Faculty of Kinesiology, Sport, and Recreation, University of Alberta, Edmonton, AB T6G 2H9, Canada; vlcarson@ualberta.ca

**Keywords:** food marketing, food environments, children, sport sponsorship, sport and recreation, public health, healthy promotion, healthy eating

## Abstract

Unhealthy food and beverage availability and sponsorship undermine healthy food practices in sport and recreation. We conducted a focused ethnography with reflexive photo-interviewing to examine parents’ awareness, reactions, and experiences of food and beverage marketing in and around their children’s physical activity in public sport and recreation facilities. Eleven parents took photos of what they thought their facility was ‘saying about food and eating’. Photos guided semi-structured interviews on the ‘4Ps’ of marketing (product, pricing, placement, promotion). Thematic analysis was conducted by holistic coding followed by in vivo, versus, and value coding. Photo-taking increased parents’ awareness of food marketing in facilities. Reactions to food and beverage marketing were positive or negative depending on parents’ perspectives of healthy food availability (choice), marketers’ motives, and mixed messages within the facility. Parents experienced their children requesting ‘junk’ food at the facility leading to parents actively attempting to reduce the frequency of these requests. Healthy eating promotion in sport and recreation facilities was misaligned with the foods and beverages available which contributed to parents’ distrust of social marketing initiatives. Critically evaluating the alignment of commercial and social marketing in recreation and sport may help inform effective healthy eating interventions that are accepted and supported by parents.

## 1. Introduction

Promoting healthy eating and physical activity are longstanding public health goals [1]. Recreation and sport facilities (RSF) are ideal spaces to encourage healthy practices among children as they often provide amenities for both play (e.g., gymnasiums, sports fields) and eating (e.g., concessions, canteens, cafes, vending machines). Research shows that parents with children who are engaged in organized sports struggle to provide healthy meals for their family due to scheduling demands of training and competition [2,3]. Home-cooked meals and snacks are often forgone, relying on convenience items, fast food meals, or foods and beverages available in vending and concessions at RSF [2,3,4].

Unfortunately, healthy eating is poorly supported in publicly funded RSF as foods and beverages available are usually considered unhealthy [5]—high in calories and nutrients of concern (e.g., total fat, saturated fat, sugar, and sodium) and low in desirable nutrients (e.g., protein, and fibre) [6]. High availability of unhealthy foods and beverages in sport settings has been demonstrated in Canada, the United States, Australia, and New Zealand [5]. Factors that contribute to unhealthy food environments in recreation and sport settings are complex and multi-level, including non-mandatory, complicated nutrition policies; competition between food services internal to RSF and food services nearby in the community; lengthy contracts with food providers that limit improvements; limited staff motivation, capacity, and resources to implement healthy food interventions; a reliance on unhealthy food sales for revenue; and social norms related to personal responsibility for health and perceived acceptability of unhealthy food and beverages in RSF [5,7,8,9].

Cultures of unhealthy eating in sport are partly facilitated by unhealthy food and beverage marketing (henceforth food marketing), such as sponsorship agreements from fast-food burger chains (e.g., McDonald’s) [5]. Unhealthy food marketing promoting energy-dense products high in fat, sugar, and/or salt has been documented in sport and recreation in Australia [10,11,12,13,14,15,16], New Zealand [17,18], and Canada [19,20]. Unhealthy food marketing in RSF is particularly prevalent in Canada, including not only sport sponsorship but also a variety of direct product, brand, and food retailer marketing through posters, branding, and product placement [19].

Unhealthy food marketing within recreation and sports leagues and facilities (i.e., clubs) has adverse impacts on product likeability and perception of health and nutrition [10,12,21,22] and may perpetuate social acceptability of unhealthy food environments in sport and recreation. A review of sponsorship in elite and community sports settings found that sport sponsorship widely exposes children and youth to unhealthy food products and brands, which influence brand awareness, preferences, and behavioural intentions [10]. For example, the majority of children at sports clubs in Australia could recall food and non-food sponsors of their local sports club and reported that they purchased food and beverage products from sponsors to ‘return the favour’ [7]. In Canada, adolescent (11–15-year-old) hockey players identified several brands and products commonly associated with the sport (Tim Horton’s^®^, Oakville, ON, CA; Subway^®,^ Milford, CT, USA; McDonald’s^®^, Chicago, IL, USA; Gatorade, Chicago, IL, USA) in a photovoice study aimed at understanding factors in recreational hockey settings that influenced youth’s eating practices [23]. Youth indicated that they felt loyal to Tim Horton’s^®^, a quick serve café with pastries, soups, and sandwiches, because of its Canadian origins and legacy of sponsoring youth hockey [23]. ‘Health halos’ are created when food and beverage brands are associated with sport through sponsorship or athlete endorsement, which may improve parents’ and children’s perceptions of the healthfulness of unhealthy products and brands [17].

Unhealthy food marketing can also undermine parents’ abilities to choose healthy foods for their children [24,25,26]. Many parents feel that the power of the food industry is overwhelming, and would support restrictions on food marketing [24,27]. Restricting unhealthy food marketing in sport and recreation facilities is recommended by the World Health Organization due to the strength of evidence that exposure to marketing for foods and non-alcoholic beverages high in fat, sugar, or salt negatively impacts children’s food-related beliefs, attitudes, preferences, and practices [28]. Limited research has explored parents’ perceptions of food environments in sport and recreation [23,29,30,31,32,33,34]. Research on parents’ perceptions of sports-related food marketing has focused on sport sponsorship and sport celebrity endorsement [30]; no research has comprehensively assessed parents’ perspectives of food marketing in RSF. Parents often play an important role in food environments in RSF by volunteering to run food services [35] for fundraising or other purposes, and thus we believe they are knowledgeable of food marketing in this setting, as well as have power to influence it.

Our study aimed to understand parents’ awareness of, reactions to, and experiences of food marketing in and around their children’s sport and physical activity in publicly funded RSF in Canada.

## 2. Materials and Methods

### 2.1. Study Design

We conducted a focused ethnography using reflexive photo interviewing—a data-generating strategy where participants’ own photographs guide conversations between the investigator and the participant [36]. Photo-based research methods have been used to understand consumer experiences of shopping at retail stores [37], advertising and marketing [36,38,39], and food and eating [40,41].

### 2.2. Participants and Recruitment

Our study was conducted following a randomized controlled trial called Eat Play Live (EPL) that evaluated the food marketing environment in RSF across Canada [42]. Of all 51 EPL RSF, we invited 11 facilities located in the province of Alberta to participate. Five RSF (45%) agreed to participate, however, two dropped out due to staff changes. The three participating RSF were publicly funded multi-sport RSF and contained 42–88 instances of observable food marketing instances per RSF, of which up to 50% were for unhealthy foods and beverages high in calories, fat, sugar, and/or sodium; associated brands; or retailers selling the same [19]. Two RSF were located in small population centres (less than 30,000 people), which each had several ice surfaces and a pool. The remaining RSF was in a large population centre (more than 100,000 people) and contained fields, courts, and gymnasiums. All three had privately run food services through concessions/franchise and vending machines.

We recruited participants through posters and in-person booths at each RSF [23]. Parents with at least one child (17 years or younger) participating in an organized physical activity at least once a week at an EPL RSF were eligible. We purposively sampled [43] parents with the aim of achieving a sample of parents who varied in age, ethnicity, gender, and socio-economic status. The sample size was driven by theoretical saturation in which sampling ceased when no concepts arose in the analysis [44]. Parents received a CAD 30 grocery store gift card for their participation.

### 2.3. Procedures

We gave interested parents an information letter and required them to sign an informed consent form before participating. The study was conducted in accordance with the Declaration of Helsinki, and the protocol was approved by the University of Alberta Human Research Ethics Board (Pro00068598). We reviewed the scope of food marketing, defined by the ‘4Ps’ [45], with parents before asking them to take photos. We explained that food marketing is broader than just advertising and that marketing can include: (a) ‘product’—what types of foods and beverages are available to purchase; (b) ‘pricing’—the costs of certain foods and beverages (financial and non-financial); (c) ‘placement’—where foods and beverages are available to purchase, how easy it is to access them, or where they are promoted; and (d) ‘promotion’—how foods and beverages are promoted through signs, messages, or programs. Participants were instructed to take any number of photos over two weeks [46] in response to the question: ‘What do you think your facility is saying about food and eating?’. Participants shared the photos that they deemed the most meaningful to them with the first author (R.P.) by email. Next, R.P. printed participant’s own photos and used them to guide individual interviews conducted at their local recreation facility the following week. See supplemental files for interview process and guide.

### 2.4. Data Generation and Analysis

RP conducted semi-structured photo-interviews in November and December 2017. The interviews started by asking participants to show and describe the photos they took, after which several topics were discussed, including:parents’ rationale for taking photos and selecting the most important ones;the ‘4Ps’ of marketing in relation to their awareness and reactions of marketing; andhow parents saw themselves, their children, and their family (i.e., their experiences) in the photos they took.

Interviews were audio-recorded, transcribed verbatim, de-identified, and analyzed using thematic analysis by R.P. [47]. All data were organized using NVivo 11 (QSR International, Burlington, MA, USA, 2017). Holistic coding was used to identify high level topics in the transcripts guided by the research question [48]. We used three coding methods (in vivo, versus, and value) designed to honor participants’ perspectives and actions were used to recode the data within each holistic code [48]. Codes were combined into sub-themes, themes, and thematic maps [47]. Themes were reviewed for internal and external homogeneity [47].

### 2.5. Rigor

Rigor was ensured by a series of verification strategies [49,50] including pre-study field engagement, ensuring methodological coherence, memoing, negative case analysis, and theoretical thinking. Our research aim was continuously reflected on to ensure methodological coherence between the research question, data generation, and data analyses [44]. Memoing [49] was actively completed during data generation and analysis to document common topics, and emerging hypotheses that were followed-up in interviews and through iterative thematic analysis. Theoretical trajectories that arose from potential negative cases (i.e., data that deviates from inductive hypotheses) [51] were investigated to understand variations in the phenomenon and revise understandings of the parents’ awareness of, reactions to, and experiences of food marketing within and across themes. Inductive and iterative theoretical thinking [49] developed a rich understanding of the phenomenon by defining single components of the phenomenon, linking components across themes, and situating the findings in the literature.

## 3. Results

Table 1 presents the sociodemographic characteristics of participants. Eleven parents participated in 10 interviews (two parents from the same household participated in one interview together). Approximately half were female, and only two self-identified as an ethnic minority. Participants were between the ages of 33 and 52 years and had between two and five children. Children of participants engaged in a variety of physical activities, with hockey, swimming, and soccer being the most common. Most (90%) visited the facility 2–5 times per week with one or more of their children. Participants submitted an average of 12 photographs. Interviews were an average of 50 min in length.

Six major themes were identified by exploring parents’ awareness of, reactions to, and experiences of food marketing in and around their children’s physical activity in their facilities: (1) raising consciousness, (2) having choice, (3) marketers’ motives, (4) mixed messages, (5) children request what they see, and (6) parents actively try to reduce children’s unhealthy food and beverage requests. See Table 2 for definitions and exemplar quotes.

### 3.1. Parents’ Awareness of Food Marketing

#### Raising Consciousness

Parents stated that they were unaware or had ‘no idea’ about food marketing before taking photos. Parents described some marketing as ‘background noise’ and attributed their lack of awareness to repeated exposure to, and the fixed nature of, food marketing in RSF (e.g., signs that have remained for years). Deliberately taking photos resulted in parents seeing new things, such as products for sale, promotional signs, product placement, and sponsor branding, and prompted them to consider the food marketing environment from different points of view (e.g., children, corporations, facility owners).

### 3.2. Parents Reactions to Food Marketing

#### 3.2.1. Having Choice

Without prompting, parents distinguished healthy and unhealthy food, not by nutritional content directly but by preparation method. Prepackaged frozen items, and foods and beverages that required little preparation were considered unhealthy. Homemade foods prepared daily with thought and care, made from scratch with fresh real ingredients, were considered healthy. Most parents believed that food and beverage options at facilities were generally not healthy; fruit and vegetables were non-existent or insufficient. Parents’ descriptions of ‘*choice*’ (used synonymously with healthy food availability) varied from ‘not a lot of choice’ to ‘nice choices’. Parents with the latter positive perspective indicated that some, albeit limited, healthier options were available: ‘I think there are choices like more healthy stuff, although it’s not 100 percent, but still it’s not bad.’ (P1); ‘They now have a butter chicken on rice. Like okay that might not be the healthiest choice, but it’s a—it’s something different.’ (P5).

Parents were generally not optimistic that they or others would choose healthy options if unhealthy options were available or promoted alongside: ‘…if there’s something that appeals to you, like some people have really good willpower and they will pick the salad nine times out of ten. But that, I think that’s pretty few and far between’ (P9); ‘…if there is the burger option, are they ever going to take the good sandwich?’ (P8B).

#### 3.2.2. Marketers’ Motives

Parents had positive and negative reactions to food marketing depending on whether they perceived marketers to be motivated to benefit people or to generate financial profit.

Marketing benefits people—Some parents believed that food marketing could help parents, their children, community business owners, or local sports. First, healthy food labelling and signage described as ‘smart’, ‘neat’, ‘cool’, and ‘fantastic’ helped parents identify and encourage healthy choices for their children. They believed healthy food promotion could guide their children’s food choices: ‘…[child] knows I can’t—she can’t have anything that doesn’t have a checkmark on it for supper’ (P10). However, not all parents had positive reactions to healthy food marketing (see Mixed Messages).

Parents believed that marketing by local businesses (e.g., through sport sponsorship) helped community business owners and local sports. In general, parents believed that local businesses who sponsored sports were altruistic. Specifically, parents believed that the primary motive behind local businesses’ decisions to provide sponsorship was truly to support local sports. Parents did not believe that the primary motive of local business sports sponsorship was to make money for their business, even if they recognized that the sponsorship could also generate revenue for the business.

Marketing generates profit—Other parents spoke more negatively about food marketing focusing on the profit-generating motives of marketers. Compared to local businesses, parents thought big business sponsors were self-serving: ‘…I like supporting the little guy, they’re just trying to put food on their table and pay their bills whereas big businesses, they’re trying to buy like mansions and stuff.’ (P5). One participant believed that while big business sponsorship of sport was financially motivated, their investment still benefitted community sports. His perspective may be unique due to familiarity with big business franchisees owners in his local community. Nevertheless, he recognized that he did not always distinguish between altruistic and profit motivated sponsors, nor did his children know the difference. This may have hindered his own or his children’s abilities to counter persuasive marketing messages. For both local and big businesses, sport sponsorship by non-food companies was seen to be more selfless (‘…trying to help the community…’ (P2)) than sponsorship by food companies (‘…trying to get more business…and get people to buy their products’ (P2)).

Financial motives of food service operators were also mentioned. Concessions and vending operators were believed to be trying to maximize profits by selling unhealthy food (while understanding that food operators may need to do so to survive) and by targeting children with the placement of unhealthy foods.

#### 3.2.3. Mixed Messages

Parents strongly felt that it did not make sense to be serving fast foods and ‘junk’ food (e.g., burgers, deep-fried foods, candy, chocolate, sugar-sweetened beverages) in RSF where the purpose was to promote active living and wellbeing. Some parents perceived the efforts to improve the food availability and marketing in facilities as insufficient or an insincere token act to give the appearance that the facility or the food service operator was committed to healthy eating: ‘…there’s the healthy eating initiative which I see that more as just lip service of okay we have to do this and put that out there, and make sure we have one option, so we can sell junk food.’ (P8A).

Parents were particularly concerned about mixed messages in that messages pertaining to food promotion and the actual availability of food within the facility were incongruent. For example, participants saw messages saying ‘eat healthy’ or ‘make better choices’ in the facility but believed there were no healthy choices to purchase. In other cases, healthy messages were placed next to an unhealthy choice. A sign saying ‘choose healthy drinks’ placed next to a machine for slushies (frozen sugar-sweetened beverages) was called ‘ironic’ and ‘disappointing’. Furthermore, some parents were wary of promotional signs or labels in the facility that they perceived health-washed unhealthy foods and beverages.

On the other hand, there were parents who thought that their own facility’s food environment was more consistent with promoting wellbeing, but they believed that other facilities presented mixed messages. The difference in perceived fit between the facility messages and actions parallel variations in the other themes of *choice* and *marketer’s motive*. Parents who believed there were no healthy choices and that marketing was primarily profit-motivated perceived more mixed messages. A better fit was perceived by parents who thought there were healthy choices and that marketing within their facility was primarily people-motivated.

### 3.3. Parents Experiences to Food Marketing

#### 3.3.1. Children Request What They See

All parents noted that their children requested ‘junk’ food in facilities and not fruits, vegetables, or water. Parents explained that their children chose what they wanted out of the products they could see since ‘what they see is what their world is’ (P8A). In particular, the enticing slushie machines located on the front counter were often mentioned: ‘you can see [slushies] as soon as you walk through the door…you can see the machine twirling (laughs), calling to children from afar.’ (P9). Participants discussed the visual presence of vending machines. When vending machines were more visible (e.g., right outside the change rooms), children requested foods or beverages more often than when vending machines were in a more discrete location (e.g., around the corner).

Parents compared colors, images, and ‘fun’ themes to written text on menus or signs. They said their children were drawn to colorful products: ‘Interviewer: Why do you think she asked for a slushie? P10:…she notices when the color changes, ‘cause sometimes it does change here…she notices immediately as soon as there’s a new flavor’. Parents did not believe that children noticed or cared about written signs and suggested that pictures were more likely than written words to impact their children’s requests.

#### 3.3.2. Parents Actively Try to Reduce Their Children’s Unhealthy Food and Beverage Requests

Parents described how they actively tried to avoid (or planned to avoid) concessions or vending machines; that they monitored, negotiated, and denied their children’s requests and choices of foods and beverages in the facility; and that they taught their children about healthier choices. Parents focused on intervening on the impact of food availability on their children’s food requests and choices rather than intervening on the impact from other aspects of marketing. For example, parents were especially apathetic about the impact of sport sponsorship arguing that their parenting was as or more impactful than promotional advertising in shaping their children’s food patterns: ‘…it’s not the advertisement that should be allowed to dictate what’s going on in my kid’s life. It’s like it should be me, and if I’m choosing a more healthy lifestyle for them, then I’m hoping my influence is more than a big red billboard on the side of arena’ (P7).

Avoid—Many participants stated that they rarely or never used (i.e., purchased foods or beverages from) the concession and/or vending machines in facilities due to perceived low availability of healthy choices, the expense of purchasing out-of-home snacks and meals, and the desire to avoid establishing a pattern of purchasing unhealthy items at the facility. Parents also stated that they avoid food services at the facility because they did not believe it meet their family values, such as eating healthily, preparing your own food, eating as a family: ‘I think for our family culture, they would know that our family values would trump what the rink offers, and so we’ve enforced at home that it’s important to eat healthy, and that doesn’t include much at the rink.’ (P3).

Plan—Planning (such as eating at home, packing snacks, and visiting nearby food retailers) was seen as necessary to ensure children were adequately fueled for their sports and to avoid using the concession and vending machines in the facility.

Monitor/negotiate/deny—Parents felt it was necessary to monitor children’s choices, believing that, if left alone, children would select unhealthy options. Parents negotiated with their children to help them identify and choose healthier options. A few parents said they denied children’s requests, which they explained as an effective strategy to curb future requests: ‘they very rarely ask me anymore…they know better…‘cause the answer’s no.’ (P5)

Teach—Some parents explained how they used the facility food environment to teach their children about healthy eating. Parents used healthy food labelling on foods and beverages in the concession and vending machines to help their children identify healthier food options available in the facility. On the other hand, one participant saw an opportunity to reinforce his family values by avoiding facility food services: ‘… it’s the opposite of what I teach them, but the fact that we don’t get anything from them is … it reinforces…what I’m teaching….’ (P2).

## 4. Discussion

Photo-interviewing served as a mechanism to raise parents’ awareness of food marketing (product, price, place, promotion) in RSF. Parents had positive and negative reactions to food marketing depending on their perspectives of food choice, marketers’ motives, and message congruency. Parents who believed that food marketing was profit-driven were unsatisfied with the availability of fresh, handmade, unprocessed healthy foods in RSF and perceived that the RSF was sending mixed messages (e.g., promoting healthy eating while selling unhealthy foods). All parents reported experiencing child requests for ‘junk’ foods (e.g., candy, slushies) while visiting RSF, stimulated by visual cues, such as product placement and eye-catching colours. As a result, parents stated they actively engaged in several strategies to reduce children’s requests.

The findings of this study have important practical and theoretical implications, each of which will be discussed in detail:Parents interpret food marketing through their understanding of ‘product’.Beliefs about marketers, whether real or perceived, influenced parents’ support for marketing activities.Understanding of food marketing in RSF, and support for healthy food marketing may be facilitated by a critical social marketing approach.

### 4.1. Practical Implication #1. Parents Interpret Food Marketing through Their Understanding of ‘Product’

Parents’ discussions of food marketing centered on ‘product’—the availability of foods and beverages—a logical focus since ‘product’ is the foundation for other marketing components [45]. Grounded in ‘product’, parents shared their perceptions of ‘promotion’, ‘placement’, and ‘price’ marketing components that embedded all themes. For example, signs, branding, sponsorship, and food labelling were mentioned in parents’ reactions to promotional messages and to marketers’ underlying motives. These promotional techniques were also discussed by parents when recounting their experiences of children’s requests and strategies to manage requests. Prominent product placement of unhealthy food options (e.g., children’s eye-level) and visibility of food services (e.g., vending machines in high traffic areas) are attributes of ‘placement’ that were perceived to be profit-motivated and to influence children’s requests. In terms of ‘price’, parents mentioned the expense of eating out as a reason to avoid food services at the facility—a strategy used to manage their experiences of food marketing. The non-financial ‘price’ of using facility food services was also mentioned, including the loss of healthy home-cooked meals and family mealtime, and inadequate ‘fuel’ for their children. Further, parents debated the financial and non-financial beneficiaries of food marketing in sport and recreation in their reactions to marketers’ motives. Parents’ interpretations of food marketing highlight the importance examining the mix of marketing strategies present in RSF and the synergy or opposition that may exist between components, especially the alignment between ‘product’ and other components. Building on parents’ focus of ‘product’, we should consider food availability as the critical component that serves as the foundation for an entire marketing strategy.

### 4.2. Practical Implication #2: Beliefs about Marketers, Whether Real or Perceived, Influenced Parents’ Support for Marketing Activities

In their reactions to food marketing, parents identified a mix of moral (for the good of society) and instrumental (for profit) motives [52] that may drive food marketers in RSF. Parents varied in their beliefs of the primary motive of marketers, which may suggest that parents, like any consumer in general, infer motives of marketers from limited information about the company they gathered from various exposures to that company [53]. Big business, especially that of food, was frequently associated with profit motives compared to local businesses. Although not explicitly stated by participants, interpretations of sponsors may have been influenced by their ‘proximity’ to the participant—parents may have inferred features of ‘closer’ (local) sponsors due to their own beliefs about their community, which could have buffered potential negative thoughts related to profit motives. Parents may not have had similar experiences or knowledge about big business sponsors and thus inferred basic profit motives common in commercial markets.

Our research showed some parents believed certain food marketing actions were ‘insincere token acts’ resulting in skepticism about motives, which can lead to distrust of firm’s actions [52]. Mixed messages from corporations can threaten relationships between the firm and customer [54,55], but a strong perceived fit between commitment to healthy eating and providing healthy options can lead to positive consumer reactions [54]. In an online experimental study of parents’ perceptions of food sponsors for children’s sport, parents thought that healthy food sponsors (i.e., brands for water, sandwich shops, whole grain high fibre cereals) and public health campaigns fit children’s sports better than unhealthy food sponsors (e.g., brands for sugar-sweetened beverages, burger/fried chicken takeaway, sugary breakfast cereals) [56]. Our study showed that parents have complex and nuanced reactions to the presence and actions of food companies and sponsors in RSF, which relate not only to the interpretation of actions by corporations (e.g., fast food restaurant sponsorship) but also internal actions taken by the RSF (e.g., promoting healthy eating messages). Policymakers and practitioners should capitalize on public perceptions of food marketing initiatives, or work to reframe public interpretations, to garner support for restricting unhealthy food marketing and promoting healthy food and beverages in children’s settings.

### 4.3. Theoretical Implication: Understanding of Food Marketing in Recreation and Sport Settings, and Support for Healthy Food Marketing May Be Facilitated by a Critical Social Marketing Approach

Critical social marketing is defined as ‘critical research from a marketing perspective on the impact commercial marketing has on society, to build the evidence base, inform upstream efforts such as advocacy, policy and regulation, and inform the development of downstream social marketing interventions’ [57]. Parents’ varied reactions may relate to the fact that parents were aware of both commercial and social marketing in the facilities. Commercial marketing is the marketing of specific goods and services [45], such as the food service establishments selling or promoting foods and beverages in the facility, or corporate sponsorship promoting food brands. On the other hand, the RSF displaying healthy eating promotional signs or adding symbols next to healthy options in concessions or vending is an example of social marketing. Social marketing uses the principles of commercial marketing (4Ps) to encourage individuals to adopt socially responsible behaviours [45]. Parents’ perceived authenticity of social marketing promoting healthy eating in the facility may have diminished when it existed adjacent to commercial food marketing that was generally unhealthy. This aligns with previous research on corporate social responsibility (CSR) where CSR initiatives, such as social marketing, philanthropy, and socially responsibly business practices [58], were not universally successful [59]. In fact, Shofner and Koo found that the ‘benefits of CSR may not apply to sporting events and sponsors when the relationship seems dishonest or is viewed as extremely incongruent’ [59] (p.2).

Positive reactions to CSR are associated with greater behavioral intentions (e.g., purchasing) [54]. In the context of our study, perceived authenticity of marketing activities influenced consumers’ support for the facility and its food services, which was evident in parents’ descriptions of their experiences—how they actively engaged in reducing their children’s request for ‘junk’ food. Parents attempted to counter food marketing in RSF by avoiding food services, denying or negotiating unhealthy food requests, and educating their children on healthy eating. Similarly, previous research showed that parents attempt to counter children’s exposure to television and in-store food marketing by restricting media use, avoiding grocery store visits with their children, shopping online, avoiding foods targeting children with giveaways, increasing their children’s media literacy, educating their children on healthy eating, and imposing rules on unacceptable foods [25]. While parents clearly explained how food availability and point-of-purchase promotions contributed to their child’s requests, they were less aware of sponsor branding and unsure how it influenced requests. As a result, parents did not discuss actions to intervene on children’s exposure to unhealthy food marketing that arose from sport sponsorship. Unmitigated exposure to sport sponsorship may be concerning since sport sponsorship is a particularly powerful marketing technique that impacts children’s brand awareness, as well as preference for, purchase of, and consumption of sponsored products [10]. A critical social marketing approach to research and practice may stimulate specific, relevant recommendations for alternative commercial marketing approaches and facility/municipal policies to improve food marketing environments, as well as generate complementary social marketing behaviour change campaigns. The success or sustainability of healthy food initiatives may be improved by understanding and aligning commercial and social marketing.

### 4.4. Limitations

The generalizability of the study findings is limited by a small sample of participating RSF and participating parents. Participating RSF were also part of the EPL study. Two of the three RSF had actively worked on improving their food environment before this study was completed. Recent changes in these RSF may have increased parents’ awareness of food marketing or generated initial reactions to food marketing, which may or may not persist over time. The sample of parents was small, and therefore we were unable to explore differences in perceptions of food marketing based on participants’ sociodemographic characteristics, including ethnicity. The parents who participated may have had more extreme positive or negative reactions to the food marketing environment than other parents in general or than parents who frequented different RSF. Due to time constraints, the data were not collected and analyzed concurrently, which may have impeded the interviewer’s ability to follow-up on emerging ideas.

Future studies may want to explore parents’ and children’s perspectives of food marketing in a wider variety of RSF with more diverse samples of parents. More research is needed to explore food sponsorship in sport and recreation in Canada, including its extent, impact on children, and its relative costs and benefits to sport and local communities.

## 5. Conclusions

Parents’ awareness of food marketing in their local RSF was low, but increased after a photo-taking activity. Reactions to food marketing in facilities varied among parents and were influenced by their perspectives of healthy food availability (choice), marketers’ motives, and perceived mixed messages with the facility. Parents actively tried to reduce children’s requests for ‘junk’ food while at RSF. Many parents perceived that the health-promoting nature of sport and recreation was misaligned with the foods and beverages that were marketed in RSF. This misalignment contributed to their distrust of healthy eating promotion initiatives, which they perceived as inauthentic.

As interest in creating healthy food environments in recreation and sport grows, researchers, practitioners, and policymakers should critically and comprehensively examine food marketing environments, including the alignment of each marketing component with healthy eating promotion. Our research showed that parents dismissed health promotion messages in RSF when they existed alongside with unhealthy food marketing, and that parents were displeased with unhealthy food marketing exposures in RSF—a setting intended to be health-promoting. The opportunity for social marketing to support healthy eating in RSF must not be undermined by competing messages from commercial marketing activities that prioritize financial gain over societal benefit.

## Figures and Tables

**Table 1 ijerph-19-02592-t001:** Sociodemographic characteristics of participants (*n* = 11).

Sociodemographic Characteristic	*n* (%)
Parents’ sex	
Male	5 (45.5%)
Female	6 (54.5%)
Parents’ age	
30–39 years	4 (36.4%)
40–49 years	6 (54.5%)
50–59 years	1 (9.1%)
Children’s age	
<2 years	1 (3.8%)
2–5 years	3 (11.5%)
6–11 years	7 (27.9%)
12–15 years	12 (46.2%)
16–17 years	2 (7.7%)
18 years and older	1 (3.8%)
Number of children ^1^	
2 children	7 (70.0%)
3 children	1 (10.0%)
4 children	1 (10.0%)
5 children	1 (10.0%)
Household income ^2^	
CAD 15,000–49,999 per year	1 (11.1%)
CAD 50,000–74,999 per year	1 (11.1%)
CAD 75,000–99,999 per year	3 (33.3%)
>CAD 100,000 per year	4 (44.4%)
Self-identified ethnic minority	2 (18.2%)

^1^ *n* = 1 missing response as 2 participants were from the same household (counted as 1). ^2^ *n* = 2 missing responses 2 participants were from the same household (counted as 1) and 1 participant declined answering.

**Table 2 ijerph-19-02592-t002:** Definitions of themes and exemplar quotes.

Theme	Exemplar Quotes
Parents’ awareness	
Raising consciousness: Parents’ level of awareness of food marketing in the facility before and after engaging in the photo interview, including the type and amount of food marketing and their explanations for level of awareness.	‘…[taking photos] made me more aware of what was going on, or at least the marketing and advertising and uh all the stuff. Like I, when you brought it up about what, what the rec center says about eating and stuff, I had no idea. Like I knew there was a concession, but I didn’t really—I didn’t really having any clue as to what it said, even though I’ve been here….we get so blind to visual advertising that it’s, especially like—especially fixed, I think. If it’s not right in your way then you just ignore it.’ (P2) ‘It just becomes background noise, everything here, ‘cause we are here so often.’ (P10)
Parents’ reactions	
Having choice: Parents’ reactions to the availability of foods and beverages for purchase at the recreation facilities.	‘Here it’s deep fried foods or popcorn, or slushes. Um, lots of Kit Kats, lots of chocolate bars, lots of pop. But not like a fruit basket, right, not a healthier choice for the children to go to…’ (P4). ‘…you can see: one, two, three, four—four shelf of pops and only two shelves for milk. So then not much of the options to choose from.’ (P6) ‘… the deep-fried list is this long, and the salad list is you know, there’s two salads to choose from’ (P9)
Marketers’ motive: Parents’ reactions to the primary motive they attributed to why food was marketed in the facility.	*Marketing benefits people* ‘I thought it was kind of neat that [food service operators] have it color coded…the reds—choose least; blue is choose sometimes; and the green is choose most often…I thought was nice, like sometimes kids, like they don’t know what’s a healthy choice, so that might help them.’ (P7) ‘…when you come to our facility and it’s like wow, you can—you don’t have to have junk, you can have anything you want really at our concession…the message I get from our rec center is that they’re trying to promote healthy eating. Um, and trying to make it easier for parents.’ (P10)
	*Marketing generates profits * ‘…some [businesses] [provide sponsorship] very selflessly, they just want to contribute, especially local businesses, they’re doing it to support local sports, to help kids get involved in something healthy for them…some of them are obviously doing it just for dollars and cents. Um. And I’m guessing the big corporations: Tim’s^1^, McDonald’s they have got that worked out to a fine science...’ (P3) ‘[food service operators] put all that fun kids’ stuff right at eye level, just like the grocery store, which is smart for them. Right, smart for the people selling it, not so good for me. Because my kids want that stuff. Right?’ (P5)
Mixed messages: Parents’ reactions to the consistency of food-, health- and nutrition-related messages within the facility.	‘it says ‘do what I say not what I do’. It’s a very inconsistent message that I see. Um, that there’s this message of eat healthy, but then they don’t necessarily put that out there and give a lot of healthy options’ (P8A) ‘we’re a healthy living facility, right, it’s mental health with the library and like keeping your brain strong; the pool; the skating; well it just to me is a hand-in-hand, right. Yeah why would you just serve poutine and burgers? [both laugh] At place where you’re trying to encourage active living.’ (P10)
Parents’ experiences	
Children request what they see: Parents’ experiences of children’s requests in the facility believed to be strongly driven by visual aspects of marketing (seeing products, colors, images).	‘…she’s going to want what she sees…she can only see what’s on the counter. So she sees slush and she see pizza um, she’s not seeing any healthy options…she’s going to pick the slush or the pizza…’ (P8A) ‘…just having things at the children’s height, right, a three year old is not going to be like oh mom I want—I want the bananas that are higher on the shelf, they’re going to just see all the pop and chips options and go for those’ (P4)
Parents actively try to reduce their children’s unhealthy food and beverage requests: Parents’ experiences of acting as gatekeepers in the facility to manage their child’s requests and diets, including avoiding (and planning to avoid) concessions and vending machines, monitoring and negotiating children’s choices, denying children’s requests, and teaching children about healthy eating.	*Avoid* ‘…it’s been a long time since we visited [a concession]…if there was vegetables or like sandwiches, or something other than chips and a slushy. I think we’d definitely consider it…. I don’t necessarily make the best choices either, so I would pick probably the same chips and I don’t want a slushy, but I’d probably get a pop… So I think, I try and avoid it so then I don’t eat that kind of stuff.’ (P9) ‘We just don’t eat concession food….It’s dino buddies and junk food that she doesn’t eat at home. [laughs]…We eat real food, we don’t eat mac and cheese, and chicken fingers, she doesn’t know what those are…we try to support the food that we want and the food that we’d have at home…’ (P8B).
	*Plan* ‘…if you plan ahead and you plan better, you can have food ready. ‘cause, I mean sometimes you’re working late and there’s not a lot of time, but you know then I usually just cook extra the night before and have stuff to grab on the way out, right...’ (P2)
	*Monitor/negotiate/deny* ‘I feel like I have to monitor what they’re getting from the vending machines. Like once in a while it’s a good treat, but if they had their way they would have a $3 treat every time we come. Right, we’re at this rink four times a week’ (P4). ‘…I’ll tell him okay don’t spend two dollars, you know, why would you spend two dollars buying those—the junk food…Or I, I gave you another two dollars and go and get a fresh juice …. I tell them the value right if you were spending two dollars, why didn’t you take another dollar or two from me and then buy something which is good for your health.’ (P1)
	*Teach* ‘we try really hard at home—my husband’s a kinesiologist, like he’s—so sport and hydration and nutrition are always really forefront in our family, so has discussions about smart choices and eating well, are always occurring.’ (P10)

## Data Availability

The data presented in this study are available on request from the corresponding author. The data are not publicly available due to participant confidentiality.

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
