# Peer review of "Choice, Motives, and Mixed Messages: A Qualitative Photo-Based Inquiry of Parents’ Perceptions of Food and Beverage Marketing to Children in Sport and Recreation Facilities"

_ijerph, 2022, doi:10.3390/ijerph19052592_

Round 1

Reviewer 1 Report

Dear Authors,

The manuscript (ijerph-1240224) submitted for the second review is now clear and easier to understand than the previous version.

The authors changed many parts of the planned paper in line with my and other reviewers' suggestions.

I would like to thank the authors for considering my comments and applaud them for the major revisions to improve their manuscript.

The abstract, discussion, and conclusion are sections that have been very well improved. The authors also corrected the references to comply with the rules of the journal.

The manuscript is interesting and valuable. Now, I am pleased to recommend the manuscript for publication in the International Journal of Environmental and Public Health without correction.

Reviewer

Author Response

Thank you for the comments! We appreciate the reviewers' time in re-reviewing the manuscript.

Reviewer 2 Report

Dear authors,

I read with interest your article “Choice, motives, and mixed messages: A qualitative photo based inquiry of parents’ perceptions of food and beverage marketing to children in sport and recreation facilities” and I think that the presented information is very interesting and useful to understand people’s perceptions of food marketing in sport facilities in Canada. Although the present form of your paper is easy to read, some lacks occur.

  • Words such as ‘unhealthy food’ and ‘healthy food’ exist throughout the manuscript. However, there are no explanations of how to define ‘healthy’ or ‘unhealthy’. Actually from food science perspective, defining healthy and unhealthy is a very hard topic – does a beef burger healthier or unhealthier than a vegetable salad with plenty of mayonnaise sources? Precise compositions of the food will have to be taken into account before any conclusions can be drawn. For processed food in our daily lives, it would be difficult to give a simple categorization. Therefore, authors will have to provide a definition/setting of healthy foods and unhealthy foods in the scenario of this manuscript. In L102, authors mentioned ‘foods high in calories, fat, sugar, and/or sodium’. Is this the criteria to distinguish healthy/unhealthy? Do you want to define ‘healthy food’ from science perspective or by consumers’ perceptions?

  • The same comment above also applies to ‘fast food’ and ‘junk food’.

  • L34-36. It is hard for readers who do not live in North America to understand what kind of foods are available in the sports centres. Why are they generally unhealthy? Any data to support the statement? Combined with the comments regarding definitions of ‘healthy’ food, I would suggest authors to further develop the introduction section in order to explain the current situation more clearly to readers.

  • I noted there is no information introducing participants’ ethnicity. Considering there are multi ethnic and cultural origins available in Canada, are samples wide enough to cover many of them? Or do samples only constrain with single/limited ethnicities? No matter which scenario it is, authors should provide this information and explain it.

  • The discussion part is relatively poor. Some interesting findings can be further dug. For example, L213-219 authors state compared with big business, local (small) business receives better feedback through sponsoring sport activities (a way of implementing CSR). Why is it? Could it be extended a bit?

Hope the above helps.

Good luck.

Reviewer 3 Report

The manuscript is a revised version based on three previous reviews. I read the paper before reading the reviewers comments and authors responses and after I also checked how authors have addressed the issues pointed by the reviewers. Below are some further recommendations:

-The study is on food and beverage marketing; thus this should be consistent all over the manuscript. In many cases it is mentioned only “food marketing”.

-Line 97-98 – were there 11 facilities invited? If so, please rephrase and avoid using ‘n’. It is confusing because n is usually the notation for the sample size and your sample is not the number of facilities invited.

-Line 116-117 – the scope of food marketing or of food and beverage marketing was presented to parents?

-How was the awareness established? It is claimed that parents had ‘no idea’ (line 179-180) but then how could they express motives about something that they had no idea (point 3.2.2)? After presenting what food marketing means how was it established whether the parents fully understood what is means?  Why was the discussion centered on the product? After the concept was exposed it would have been expected to be focused on all 4Ps, especially in an interview.

-Please review the discussion sections. It seems to be some inconsistencies. For instance, parents centered on ‘product’ is one statement and later it is pointed “highlighting the importance of examining the collective mix of marketing strategies present in sport and recreation facilities”

-Line 405-421 – please rephrase. It is more precise to start by saying that the low number of facilities and the low number parents as participants reduces the generalization. And after explain the situation for both, the facilities and the parents.

-I was also wondering whether all facilitates provide same services, at same prices, are placed in same type of environment etc.

-And not the least are the practical and theoretical implications of the study.

Round 2

Reviewer 3 Report

The manuscript is improved based on the recommendations. It is an interesting research. Adding the supplementary file clarifies the steps taken.

This manuscript is a resubmission of an earlier submission. The following is a list of the peer review reports and author responses from that submission.

Round 1

Reviewer 1 Report

This study provides a novel insight regarding parents’ perceptions of food marketing in their children’s sport and rec facilities. I have a few considerations for the authors. Introduction:

Page 1, lines 38-40 – Can you be more specific here? Are these time constraints and resulting fast food consumption noted specifically on training/competition days? Or is this in general? Can any other similar evidence be brought in here – is this association commonly reported or only in this one study?

Page 2, lines 52-54 – Again, can you be more specific here about the association seen between food marking at PA/sport venues and the harmful impacts? Is it that unhealthy food marking within sport/PA is associated with children liking those unhealthy foods and having a poor perception of what is healthy and nutritious? The last sentence in this paragraph is really clear and succinct.

Page 2, lines 63-64 – Include reference to children in this sentence – i.e. limited research has explored parents’ perceptions of the food environments in children’s sport and recreation.

Page 2, lines 65-67 – perhaps this is also community/club-sport where the research is lacking? Materials and methods: Language is largely in third person, however this paragraph begins with ‘we conducted….’. Change to third person to be consistent with the rest of the manuscript. Please indicate whether ethical approval was received prior to undertaking this project.

Page 3, lines 114-115 – This sentence is incomplete. Please edit. Is it that each participant’s photos were printed and used at their photo-interview which was conducted at their rec facility? Results: Table 2 – Rather than just the participant number, can you provide an indication of parent sex, age, and whether self-identified as ethnic minority for each of the exemplar quotes? This is really important as we know mothers and fathers differ in their food provision, and there are also likely to be differences according to ethic background. Although the sample size is too small to make inferences based on demographic characteristic differences, it can still provide insight into potential differences. It would also be good to indicate whether the participants attended the facilities that had improved their food environment or not (acknowledged as a limitation).

Page 8, lines 264-265 – This statement refers to Figure 1, yet the Figure is labelled ‘Figure 2’. Please correct the Figure label.

Reviewer 2 Report

Dear Authors,

The manuscript (ijerph-1240224) presented for review is very interesting but I have mixed feelings in relation to the quality of this manuscript. I understand that is qualitative research, but it doesn’t convince me.

The authors agree with my opinion in section Limitations.

First, the group of children parents accounts only 11 people, and moreover, two people are from the same family (parents).

Is it possible to conclude so widely about food marketing in sport and recreation-based only on opinion 11 of random people?

The authors did not take into account the parents' education at all, which could have significantly influenced the results of the study. It is true that the people participating in the study had different financial statuses, but they were 1-2 people of different economic statuses. In my opinion, the study was not properly conducted.

The aim of this study is to understand food marketing in sport and recreation facilities in Canada from the perspective of parents of children who regularly visit their local publicly funded sport and recreation facility for physical activities. In my opinion in the section Conclusion authors did not properly respond to the set aim.

The results obtained are not presented clearly and it is difficult to understand what the authors actually found in their research and what it means.

The list of References in positions 14 and 53 is only BLINDED – what is it?

Reviewer 3 Report

This study aimed to understand the state of food marketing in sport and recreation facilities in Canada from the perspective of parents of children who regularly visit their local publicly funded sport and recreation facility for physical activities. The study is relevant for this journal as it contributes to issues related to public health. Still the discussion is very poor as results continue to be repeated in several sections. It originally focused on marketing, still results only focused on product, either the parents did not understand the leading questions or it were not properly informed, still this is a main issue for this paper. More comments follow.

L32 p1. Missing reference.

L68-70 p2. It might be better not to quote the original research. The first section is redundant, it has been said above and the consideration of whom must give an opinion could be rephrase by the authors, so they could establish and emphasis on their own research perspective.

L77-79 p2. As they had established the objective of the study just above, here the question results redundant.

L51 p2. Reference 14 is BLINDED, this couldn’t be verified by the reviewer.  Same as Ref 37 and 53. Please explain.

L113 p3. Rephrase sentence.

L130 p3 Did the authors meant consistency?

L132 p2 The reference does not cover the rigor verification strategies they enlisted, more references should be added. In the following sentences verification strategies are explained, therefore this sentences could be removed.

L134-136 p2 No evidence supports this comment, please provide evidence.

L141-144 p2 Reference is missing. As for most of the lines of point 2.5.

L146-152 p4 as well as Table 1 could be better placed in the materials and methods section

P9 & P10. Figure 2, is not really clear, I suggest to be removed. A description in the text should be enough.

L271-289 p9. Results are repeated

L321-324 p10. Discussion seems to be made from the researchers’ point of view so that no data from the study seems to support it. The section should be reviewed.

 L367-370 p11. This seems as a conclusion rather than a discussion.

L387 p11. Researchers talk about food marketing, still they acknowledge that they mainly focused on the product, please reconsider using the correct term.

L394 p 12 Results are repeated